

# Shared mycorrhizae but distinct communities of other root-associated microbes on co-occurring native and invasive maples

Tonia DeBellis[1,2], Steven W. Kembel[3] and Jean-Philippe Lessard[1]

[1] Department of Biology, Concordia University, Montreal, Quebec, Canada
[2] Department of Biology, Dawson College, Montreal, Quebec, Canada
[3] Département des sciences Biologiques, Université du Québec à Montréal, Montréal, Québec, Canada

Corresponding author
Tonia DeBellis,
tdebellis@dawsoncollege.qc.ca

## ABSTRACT

**Background**. Biological invasions are major drivers of environmental change that can significantly alter ecosystem function and diversity. In plants, soil microbes play an important role in plant establishment and growth; however, relatively little is known about the role they might play in biological invasions. A first step to assess whether root microbes may be playing a role in the invasion process is to find out if invasive plants host different microbes than neighbouring native plant species.

**Methods**. In this study we investigated differences in root associated microbes of native sugar maple (*Acer saccharum* Marsh.) and exotic Norway maple (*A. platanoides* L.) collected from a forested reserve in eastern Canada. We used microscopy to examine root fungi and high-throughput sequencing to characterize the bacterial, fungal and arbuscular mycorrhizal communities of both maple species over one growing season.

**Results**. We found differences in root associated bacterial and fungal communities between host species. Norway maple had a higher bacterial and fungal OTU (operational taxonomic units) richness compared to sugar maple, and the indicator species analysis revealed that nine fungal OTUs and three bacterial OTUs had a significant preference for sugar maple. The dominant bacterial phyla found on the roots of both maple species were Actinobacteria and Proteobacteria. The most common fungal orders associated with the Norway maple roots (in descending order) were Helotiales, Agaricales, Pleosporales, Hypocreales, Trechisporales while the Agaricales, Pleosporales, Helotiales, Capnodiales and Hypocreales were the dominant orders present in the sugar maple roots. Dark septate fungi colonization levels were higher in the sugar maple, but no differences in arbuscular mycorrhizal fungal communities and colonization rates were detected between maple species.

**Discussion**. Our findings show that two congeneric plant species grown in close proximity can harbor distinct root microbial communities. These findings provide further support for the importance of plant species in structuring root associated microbe communities. The high colonization levels observed in Norway maple demonstrates its compatibility with arbuscular mycorrhizal fungi in the introduced range. Plant-associated microbial communities can affect host fitness and function in many ways; therefore, the observed differences suggest a possibility that biotic interactions can influence the dynamics between native and invasive species.

## INTRODUCTION

Invasion of native ecosystems by non-native species can alter the structure of biological communities, often leading to changes in biodiversity (*Thomas & Palmer, 2015*; *Bellard, Cassey & Blackburn, 2016*) or ecosystem functioning (*Ehrenfeld, 2010*; *Vilà et al., 2011*). Studies attempting to determine what factors may be playing a role in the invasion process have shown that multiple factors such as facilitation, enemy release, competitive release and propagule pressure can simultaneously influence invasion success (*Lockwood, Hoopes & Marchetti, 2007*). Invasions of trees and shrubs in the last few decades around the world has revealed that woody plants are among the most widespread of invasive organisms (*Richardson & Rejmánek, 2011*). Natural plant populations associate with a multitude of soil microbes and several hypotheses aimed at explaining the success of exotic plants revolve around the biotic interactions between plants and their associated microbial communities. Feedback between plants and root-associated microbes can be positive when mycorrhizal fungi and N-fixing bacteria are involved, or negative when the plant associates with microbial pathogens and parasites (*Bever, 2003*). Therefore, microbial communities in soil can influence invasion success of plants in several ways (*Wolfe & Klironomos, 2005*; *Reinhart & Callaway, 2006*; *Pringle et al., 2009*). Nevertheless, how soil microbial communities differ between native and exotic plant species remains largely unexplored. Here, we investigate differences in the microbes associated with the roots of native sugar maple and exotic Norway maple from a forested reserve in eastern Canada.

Several mechanisms directly or indirectly involving soil microorganisms are proposed to explain the success of invasive plants in their introduced ranges. The two functional groups of microbes hypothesized to have strong effects on invasive plants are the pathogens and mutualistic symbionts due to their important influence on plant growth and performance. Pathogens that are present in the native range may not occur in the introduced range such that exotic species gain a competitive advantage over native species (*Callaway et al., 2004*; *Reinhart et al., 2010*). This phenomenon is consistent with the enemy release hypothesis, which posits that invasive species thrive in their new range by no longer being hindered by their natural enemies (*Elton, 1958*; *Keane & Crawley, 2002*). For plants that require soil-borne mutualists, various strategies facilitating invasion have been proposed. Plants will not be hindered in their new range if they are able to associate with novel mutualists, associate with generalist symbionts, or co-invade with their native symbionts (*Nuñez & Dickie, 2014*). Although the role of soil microbes in the invasion process of plants is becoming increasingly recognized (*Wolfe & Klironomos, 2005*; *Reinhart & Callaway, 2006*; *Inderjit & Van der Putten, 2010*; *Dickie et al., 2017*), we still lack an understanding of these complex dynamics for most introduced species.

Plants and soil microbes form complex interactions which can affect host fitness and function (*Vandenkoornhuyse et al., 2015*). Many bacteria are known to be able to stimulate plant growth through direct or indirect interactions with plant roots. Root associated

bacteria may contribute to plant nutrition by nutrient solubilization or nitrogen fixation; to plant growth and stress release by phytohormone production or degradation, and to pathogen suppression (*Berg et al., 2016*). Soil fungi also assist plants with nutrient availability, protection against pathogens and stress tolerance (*Rodriguez & Redman, 2008*; *Van Wees, Van der Ent & Pieterse, 2008*; *Vandenkoornhuyse et al., 2015*). Soil bacteria and fungi can also be pathogenic to plants and can therefore affect the distribution and abundance of plant species by suppressing plant recruitment, growth and survival (*Packer & Clay, 2000*; *Bever, 2003*; *Reinhart et al., 2003*). An understanding of these complex relationships requires an assessment of the resident microbes on different plant hosts.

Plant species identity can be an important factor influencing root microbial diversity; however, this relationship is not universal. It has been shown to be a significant factor in some field studies (*Aleklett et al., 2015*; *Uroz et al., 2016*) and not significant in others (*Knapp, Pintye & Kovács, 2012*; *Johansen et al., 2016*). The phylogenetic relationship among plant hosts being compared may be an important factor affecting plant-microbe associations. Related plants tend to have more similar microbial communities because host traits that influence interactions with microbes tend to be phylogenetically conserved (*Wehner et al., 2014*). *Laforest-Lapointe, Messier & Kembel (2016)* examined bacterial communities of five temperate tree species and found that red maple (*A. rubrum* L.) and sugar maple have similar bacterial communities when compared to paper birch (*Betula papyrifera* Marsh.) and to the two coniferous species sampled. However, other studies have reported the opposite, with closely related plant species hosting distinct root microbes (*Reinhart & Anacker, 2014*; *Veresoglou & Rillig, 2014*). Hence, a first step to infer the role of soil microbes in the invasion process is to discover if there are different communities of microbes around two congeneric plant species growing in close vicinity.

Norway maple is an invasive tree species found in temperate forests of the northeastern United States and Canada. It is a European native that was introduced to North America around 1756 (*Nowak & Rowntree, 1990*). It became a popular choice for street trees after WWII when the native white elm population (*Ulmus americanum* L.) was devastated by Dutch Elm disease (*Nowak & Rowntree, 1990*), but it is now invading wild woodland and natural areas in urban settings. Previous studies attributed the success of the Norway maple in its introduced range to its fast growth, high shade tolerance, high seed output, and tolerance to a wide range of environmental conditions (*Kloeppel & Abrams, 1995*; *Wyckoff & Webb, 1996*; *Webb et al., 2000*; *Meiners, 2005*). Many studies have reported reduced species richness of native species and increased abundances of Norway maple seedlings relative to nearby non-invaded areas (*Wyckoff & Webb, 1996*; *Martin, 1999*; *Reinhart, Greene & Callaway, 2005*). Norway maple can inhibit the growth of native species (*Reinhart, Greene & Callaway, 2005*; *Galbraith-Kent & Handel, 2008*) including the native sugar maple by outcompeting it (*Wyckoff & Webb, 1996*; *Martin, 1999*). *Reinhart & Callaway (2004)* have shown that Norway maple can experience increased growth when grown in soil from its invaded range, and another experimental study showed that Norway maple had significantly greater aboveground and root biomass than sugar maple (*Paquette et al., 2012*). However, no study to date has investigated whether there are differences in

root microbial communities associated with native and introduced maples using amplicon sequencing of DNA extracted directly from root tissue.

The roots of maple trees (*Acer* spp.) are colonized by arbuscular mycorrhizal fungi (hereafter AMF), which are symbiotic micro-organisms that colonize the roots of the majority of the world's land plant species (*Smith & Read, 2008*). AMF provide host plants with nutrients, water, heavy metal tolerance, and/or enhanced pathogen resistance in exchange for carbon (*Smith & Read, 2008*) and could therefore promote the establishment and spread of invasive plant species (*Marler et al., 1999*). A meta-analysis of 67 studies found that the degree of AMF colonization (measured as a percentage) of invasive plants did not differ from that of native species across studies, but AMF communities associated with co-occurring invasive and native plants can differ substantially (*Bunn, Ramsey & Lekberg, 2015*). Different AMF fungi have differing effects on their host (*Klironomos, 2003*), therefore changes in AMF communities may impact plant performance and competitiveness. Studies show that exotic plants are primarily colonized with dominant, widespread AMF rather than rare AMF taxa (*Moora et al., 2011*; *Bunn et al., 2014*). Most AMF have low levels of endemism (*Davison et al., 2015*), therefore these widespread AMF with low species specificity may have the ability to quickly colonize alien hosts, possibly aiding the invasion process (*Moora et al., 2011*; *Dickie et al., 2017*). Beyond overall root colonization levels, another method used to further examine the state of the AMF symbiosis is to examine the arbuscule to vesicle ratio. Arbuscules are the site of nutrient exchange in AMF, while vesicles are fungal storage structures (*Smith & Read, 2008*), therefore, high arbuscule numbers coupled with low vesicle numbers is usually a sign of a healthy mycorrhizal association. A decreased number of arbuscules with higher vesicle numbers have been observed in the roots of stressed sugar maples (*Cooke, Widden & Halloran, 1992*; *Klironomos, 1995*). A higher arbuscule to vesicle ratio can reveal another competitive advantage in co-occurring plant species.

Here, we examined the microbial communities associated with the roots of co-occurring non-native Norway maple and native sugar maple from a temperate forested reserve. We used microscopy and DNA sequencing-based methods targeting bacteria, fungi and AMF in maple roots to address the following questions: (1) Do root-associated microbial communities vary in their composition between co-occurring native and exotic maple seedlings? (2) Do root fungal colonization rates differ between the two species? (3) Are the arbuscule to vesicle ratios higher in the Norway maple?

## MATERIALS & METHODS
### Study site and root collection
Maple seedlings were collected from the Morgan Arboretum, a forested reserve located in Sainte-Anne-de-Bellevue, Québec, Canada (45.43N, 73.94W). Norway maple trees were planted in the Arboretum in the mid-1950s, and these mature trees were cut and removed from the site in 2008. Although mature trees have been removed, many seedlings are still present in the reserve. In an area containing many Norway and sugar maple seedlings, five parallel 60 m transects (separated by 10 m) were set up. Along each transect, we randomly

selected a sampling point. On each sampling date, one Norway maple and one sugar maple seedling of similar height (~25 cm) were sampled within a radius of 1.5m for each transect. A total of 10 seedlings were collected on each sampling date. Seedlings were selected because entire seedlings could be collected in the field ensuring the exact identification of each root system examined. We repeated this protocol eight times at intervals of two weeks between June and October 2015. A total of 80 seedlings were collected (5 transects × 8 dates × 2 species). Each seedling was removed from the soil with a trowel, leaving soil around the roots to prevent desiccation. Seedlings were then placed in sealable plastic bags and stored in a cooler with ice packs until they were returned to the laboratory for further processing.

Once in the laboratory, roots were carefully removed from the soil, were thoroughly washed with tap water and the small pale feeder roots were isolated. Half of the roots were preserved in Formalin-Acetic Acid-Alcohol (FAA) solution at room temperature for a minimum of 24 h for the morphological analysis and the other half were dried using paper towels and placed in 1.5 ml microtubes and stored at −80 °C for the molecular analysis of the root microbial community. While collecting the roots, approximately 100 ml of soil surrounding the roots of each seedling was collected, air-dried for 48 hrs and stored in paper bags for soil chemical analysis (see Supplemental Information for details of the soil chemical analyses). Seedling collection and root processing until this point occurred on the same day.

## DNA Extraction

DNA was extracted from approximately 150 mg of root sample using the PowerSoil DNA Isolation Kit (MOBIO Laboratories, Inc., Carlsbad, CA, USA) with the two following modifications. Firstly, in order to increase cell disruption and increase DNA yield, samples were milled using a MiniBead Beadbeater-16 (BioSpec Products, Bartlesville, OK, USA) for 2 min with three 2.3 mm diameter stainless steel beads (BioSpec Products). After this step, the roots were paste-like in consistency therefore the contents of the PowerBead Tubes tube were added to the tubes containing the crushed roots and the procedure was continued following manufacturer's instructions. The second modification occurred at the final elution step, where 75 µl of solution C6 solution was added to the spin filter tube instead of the recommended 100 µl, also an effort to increase final DNA concentration. Extracted DNA was stored at −80 °C.

## DNA library preparation and sequencing

For bacteria, we amplified the V5-V6 region of the bacterial 16S ribosomal RNA gene using the chloroplast-excluding primers 799F-1115R (799F: AACMGGATTAGATACCCKG; 1115R: AGGGTTGCGCTCGTTG; *Chelius & Triplett, 2001*; *Redford et al., 2010*) in order to eliminate contamination by host plant DNA. For the fungal communities, the internal transcribed spacer (ITS) region was amplified using the fungal specific primers ITS-1F (CTTGGTCATTTAGAGGAAGTAA; *Gardes & Bruns, 1993*) and ITS2 (GCT-GCGTTCTTCATCGATGC; *White et al., 1990*). For the AMF, we used the Glomeromycota specific primer AML2 (GAACCCAAACACTTTGGTTTCC; *Lee, Lee & Young, 2008* and the universal eukaryotic primer WANDA (CAGCCGCGGTAATTCCAGCT; *Dumbrell et*

*al., 2011*). The WANDA and AML2 primer pair amplifies a portion of the small subunit (SSU) RNA gene. We chose this region because it is frequently targeted, and the curated MaarjAM database (*Öpik et al., 2010*) has been developed based on this region, allowing us to compare our sequences to the largest AMF sequence database to date.

The forward and reverse primers used in the PCR consisted of the target primer, a unique twelve-nucleotide barcode, and an Illumina adaptor (5′ –Illumina adaptor –12 nucleotide barcode –target primer). Reactions were performed using 0.5 unit of Phusion Hot Start II High-Fidelity DNA polymerase and accompanying 5X Phusion HF Buffer (Thermo Scientific) with one μL of undiluted root DNA for bacteria and fungi but a 1/10 DNA dilution was used for the AMF. The reactions were carried out in a final volume of 25 μl in the presence of 200 μM dNTPs, 200 pmols of each primer, and 0.75 μl of DMSO. The reaction was run on an Eppendorf MasterCycler thermocycler with a 30 s initial denaturation step at 98 °C and 35 cycles of 98 °C for 15 s, annealing at 64 °C for 30 s, and extension at 72 °C for 30 s, with a final extension at 72 °C for 10 min. Controls with no DNA and positives with target DNA were run with every series of amplifications to test for the presence of contaminants and ensure PCR reaction mix was correct. The resulting PCR products were loaded on a 1% agarose gel to verify each PCR run. All amplified samples were cleaned and normalized using the Invitrogen Sequalprep PCR Cleanup and Normalization Kit. Multiplexed amplicon libraries for each of the 3 groups were prepared by mixing equimolar concentrations of DNA, and each library was quantified using Qubit dsDNA High Sensitivity kit. Each DNA library was sequenced using an Illumina MiSeq Sequencer with the 2× 300 bp paired end platform using the Illunima MiSeq Reagent Kit v3.

## Bioinformatics

Illumina adapters were trimmed from the raw sequences using BBDuk (BBtools package, https://jgi.doe.gov/data-and-tools/bbtools/), and the trimmed paired-end sequences were assembled with PEAR (*Zhang et al., 2014*). Sequences were demultiplexed using default settings in QIIME (v. 1.9.1) (*Caporaso et al., 2010*) with the exception of allowing for two primer mismatches. Chimeric sequences were identified using the Usearch algorithm and then eliminated. We then binned the remaining sequences into operational taxonomic units (OTUs) at a 97% sequence similarity cutoff. Taxonomic identity of each OTU was determined with QIIME using the uclust algorithm and the Greengenes database ver 13_8 (*DeSantis et al., 2006*) for bacteria; RDP classifier and the UNITE database ver 12_11 (*Kõljalg et al., 2005*) for fungi, and the uclust algorithm and MaarjAM database for AMF (*Öpik et al., 2010*).

## Root staining and quantification of AMF and Dark Septate Endophyte colonization in maple roots

The preserved roots were removed from the FAA solution and placed separately in OmniSette® Tissue-Teks® (Fisher Scientific). Samples were then cleared by autoclaving for 20 min in 10% KOH. The autoclave step was repeated two times, changing the KOH solution each time. If the KOH solution was a dark brown color after the second autoclave step the process was repeated a third time in order to ensure the roots would be sufficiently

cleared for morphological analysis. Following the KOH step, roots were gently rinsed with tap water and bleached with 35% hydrogen peroxide for 30 min. Samples were then rinsed again with tap water and acidified in 15% HCl for 15 min. Finally, roots were stained in 0.1% chlorazol black E at 90 °C for 12 min (*Brundrett, Piché & Peterson, 1984*). After staining, the samples were allowed to destain in a 50% glycerin solution overnight. The roots were then mounted on slides in glycerine jelly and squashed with a cover slip (*Widden, 2001*). Root fungi were examined using a Leica DM6000 light microscope at a magnification of 400x. The colonization rate for each root sample was quantified using the magnified grid-intersect method (*McGonigle et al., 1990*). Each intersect was evaluated for the following AMF structures: hyphae, arbuscules and vesicles, for a total of 100 intersects per root sample. For each fungal structure, the colonization rate was determined by counting the total number of intersects in which it was present. Total AMF colonization rates were determined by counting all intersects in which at least one AMF structure was present. When the stained roots were observed under the microscope, the dark septate endophytic (DSE) fungi were clearly evident and easily distinguishable from the AMF, therefore at each intersect the presence of these fungi were also noted and quantified.

## Statistical analysis

Prior to the analyses of the microbial sequence data, in order to eliminate potentially spurious OTUs caused by PCR error or sequencing artifacts, all OTUs with fewer than 10 occurrences were removed and the number of sequences per microbial community was rarefied based on the number of sequences present in the sample with the lowest number sequences in order to maintain equal sampling depth across samples. To illustrate differences in the structure of bacterial and fungal communities, non-metric multidimensional scaling (NMDS) analysis based on Bray–Curtis similarities at the OTU level (using Hellinger transformation) was used, and the significance of the observed differences were determined by PERMANOVA (distance-based permutational multivariable analysis of variance, *Anderson, 2001*) using 999 permutations. To identify which OTUs differed in abundance between maple species we used ANCOM (Analysis of composition of microbiomes; (*Mandal et al., 2015*) implemented in R, with the multcorr2 parameter. ANCOM was used because of its power and accuracy in the detection of differential abundance of OTUs and for its low false detection rate (*Weiss et al., 2017*). For this analysis, unrarefied community datasets are used, and all OTUs with fewer than 10 sequence reads were removed. We also classified the functional guild of the 25 most common fungal OTUs using FUNGuild (*Nguyen et al., 2016*). These 25 OTUs included all OTUs that had >0.5% relative abundance for each maple species. In brief, FUNGuild uses the given OTU taxonomy assigned file to compare it to its database to attempt to assign an ecological guild to each OTU. For each match a confidence rating is also given with highly probable being absolutely certain, probable being fairly certain and the lowest rating being possible, meaning the identification is suspected but not highly supported.

We calculated richness of all three microbe groups from the OTU community matrices and performed $t$-tests to test for differences between species. The overall effects of species, date and interaction between species and date on AMF and DSE colonization were

determined with two-way ANOVAs. Two-way ANOVAs were also used to determine if there were any significant differences in soil chemical parameters with species and sampling date. When data did not satisfy the Shapiro–Wilk test for normality a log transformation was performed. All statistical analyses and visualization were performed in R (*R Core Team, 2013*) using the following packages: ggplot2 (*Wickham, 2009*), picante (*Kembel et al., 2010*), vegan (*Oksanen et al., 2017*), and ANCOM (*Mandal et al., 2015*).

The metadata and sequences files have been deposited in Figshare (see DNA Deposition and Data Availability sections). Raw sequence reads have also been deposited in the NCBI Sequence Read Archive under accession number PRJNA553535.

## RESULTS

### Taxonomic identification of soil microbial communities associated with the roots of Norway and sugar maples

Due to a contamination with one of the forward primers used, all 10 samples from the June sampling date were removed from the analysis and we did not obtain any sequences from one Norway maple sampled on July 11th. From the remaining 69 samples, we obtained a total of 207,280 bacterial sequences after quality trimming and the removal of chimeras. Once all OTUs with fewer than 10 sequences were removed, and the dataset was rarefied to 1,000 sequences per sample, a total of 66 samples and 1077 OTUs remained and were used in the analyses. A total of 18 bacterial phyla were detected from the maple roots, and these 18 phyla were further classified into 47 bacterial orders with 12 having an abundance of over 1% (Fig. 1A). Approximately 16% of the OTUs were not identified to order.

A total 256 389 fungal (ITS) sequences were obtained after quality trimming and the removal of chimeras. When all OTUs with fewer than 10 sequences were removed, and the dataset was rarefied to 1,000 sequences per sample, a total of 75 samples were included in the analyses and the remaining sequences clustered into 474 OTUs. The majority of OTUs were Basidiomycota 50.2%, followed by Ascomycota 31.1%, Glomeromycota 4.3%, Zygomycota 1.7% and 12.7% were unidentified at phylum level. The OTUs were classified into 40 orders, with 11 having a relative abundance of over 1% (Fig. 1B).

Using the AMF specific primers, we obtained sequences from 79 of the 80 samples. A total of 166 403 sequences were obtained after quality trimming and the removal of chimeras. Once all OTUs with fewer than 10 sequences were removed, and the dataset was rarefied 800, a total of 76 samples were included in the analyses and the remaining sequences clustered into 199 OTUs. The relative abundance of the AMF OTUs classified to family level was similar for both host species (Fig. 1C).

### Quantification of differences in the composition of soil microbial communities associated with the roots of Norway and sugar maples

We found that the composition of bacterial and fungal communities associated with the roots of Norway and sugar maple were distinct from each other, while the AMF communities were not distinguishable based on plant host (Fig. 2, Table 1). Host species explained 3.6% of the variation in root bacterial and 8.4% of the variation for the fungal community structure. Sampling date explained 3.6% of the total variation in the bacterial

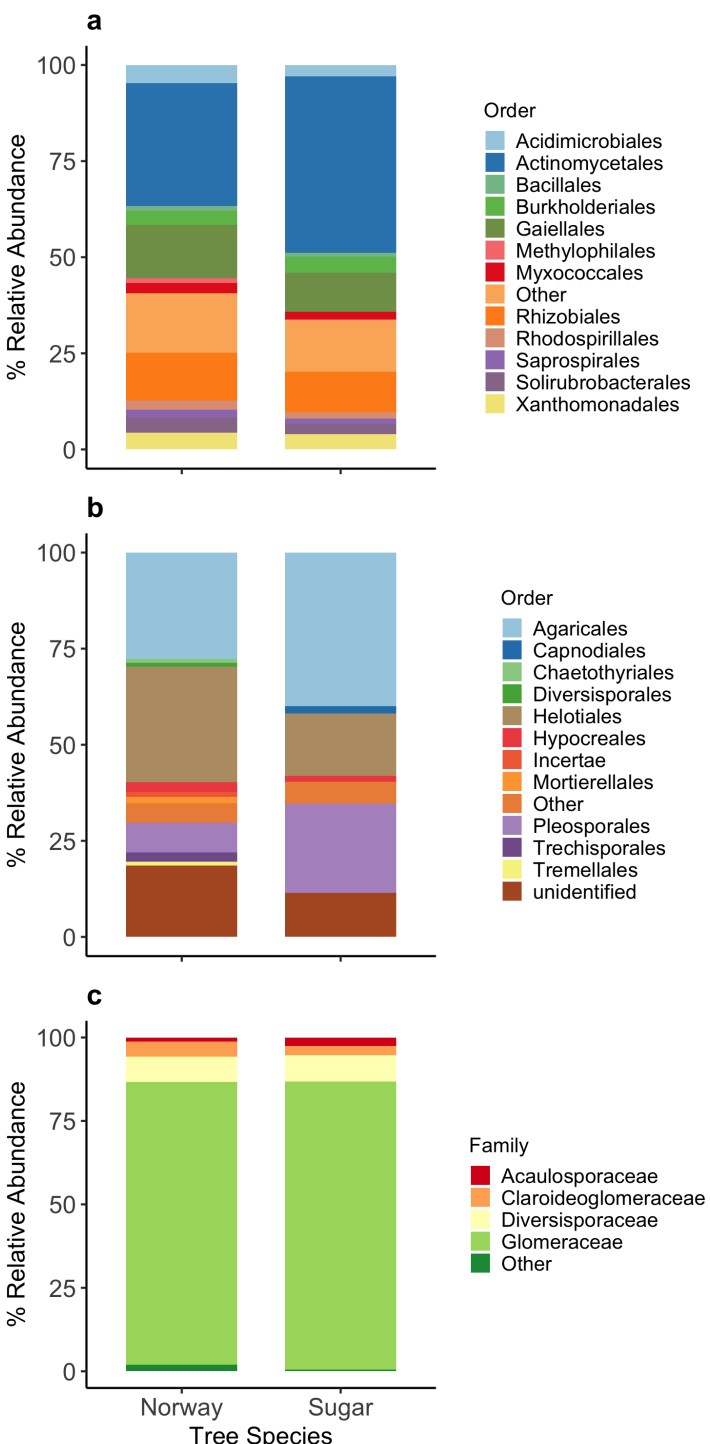

**Figure 1 Relative abundance (%) of the main taxa detected among the microbial communities in the roots of Norway and sugar maple.** (A) Bacterial communities (Order); (B) Fungi ITS (Order); (C) AMF (Family level). The stacked bar graphs represent the overall relative abundance across the entire dataset.

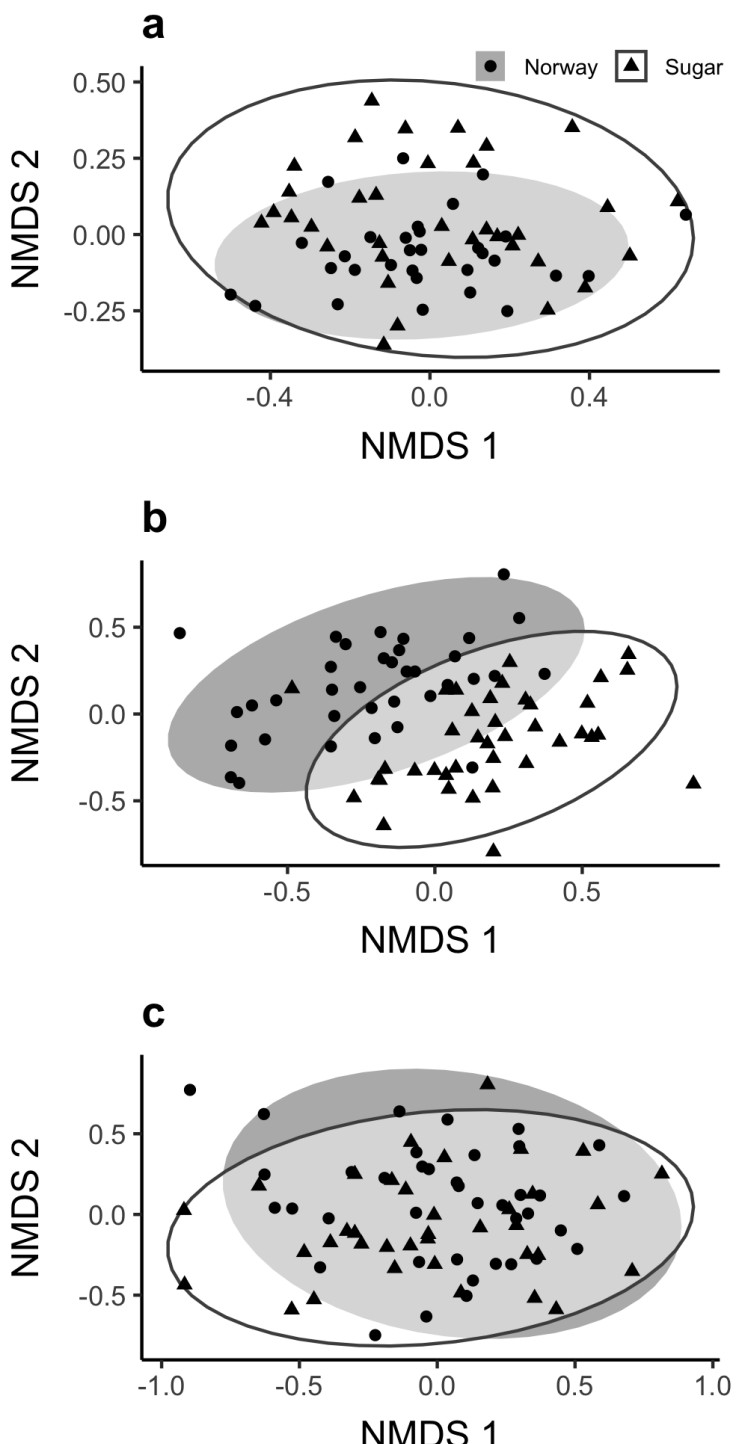

**Figure 2** **Plots of (A) Bacterial, (B) Fungal, and (C) AMF communities associated with the roots of sugar and Norway maples based on 16S, ITS and 18S regions (respectively) using NMDS scaling.** Ordinations based on Bray–Curtis distances among samples. Ellipses indicate 2 standard deviations around samples from each host species. Stress values for bacterial, fungal and AMF ordinations are 0.17 in 2D space, 0.17 and 0.19 in 3D space, respectively.

**Table 1 Microbial community structure variation among maple roots explained by various factors (PERMANOVA on Bray–Curtis dissimilarities).** *P*-values are based on 999 permutations.

| | | Bray–Curtis dissimilarities | |
|---|---|---|---|
| Microbial Community | Variable | $R^2$ (%) | Pr (>F) |
| Bacteria | Species | 3.6 | .001 |
| | Date | 3.6 | .003 |
| | Species*Date | 1.7 | .204 |
| Fungi | Species | 8.4 | .001 |
| | Date | 1.8 | .084 |
| | Species*Date | 1.2 | .519 |
| AMF | Species | 1.5 | .349 |
| | Date | 0.6 | .979 |
| | Species*Date | 1.2 | .576 |

communities and had no significant effect on the variation in the fungal communities (Table 1).

We also found significant differences in the relative abundance of particular OTUs between the native and invasive species. Many OTUs from each microbial group were found in both maple species, with plant hosts sharing 95.4%, 65.6% and 83% of all bacterial, fungal and AMF OTUs, respectively. However, when we compared OTU abundance between species using ANCOM (*Mandal et al., 2015*), we found three bacterial and 10 fungal (ITS) OTUs whose abundances differed significantly between the sugar and Norway maples ($P < 0.05$, Fig. 3). All three bacterial OTUs and nine of the 10 fungal OTUs had significantly higher abundance in sugar maple compared to Norway maple (Fig. 3). The ten fungal OTUs that varied significantly in abundance between maple species, represent 12.1% of the relative sequence abundance in Norway maple and 39.7% of the sequences in sugar maple. The nine fungal OTUs that were found in significantly greater abundances in sugar maple accounted for 37% of its total OTU relative abundance compared to only 6.6% in Norway maple. The three differentially abundant bacterial OTUs represent 7.2% and 15.0% of the relative sequences in Norway maple and sugar maple, respectively.

Only 8 out of the 25 most commonly occurring OTUs could be classified to a functional group using FUNGuild, with the remaining left unidentified. These 25 OTUs made up 74% of the relative OTU abundance in Norway maple and 82% in sugar maple. Only two OTU functional guild classifications were ranked as highly probable and both were categorized as symbiotic ectomycorrhizal fungi. Four OTUs were ranked as probable and two possible. These six OTUs were either classified as solely saprotrophs or another category which included saprotrophs, symbionts and pathogens. The top two fungal OTUs with higher abundances in sugar maple (Fig. 3, Didymosphaeriaceae and *Entoloma* sp), were classified as saprotroph-symbiont-pathogen with Didymosphaeriaceae having the 'probable' rating and *Entoloma* sp. 'possible'. These two OTUs made up 30.8% and 5.2% of the OTU relative abundance in sugar and Norway maple, respectively.

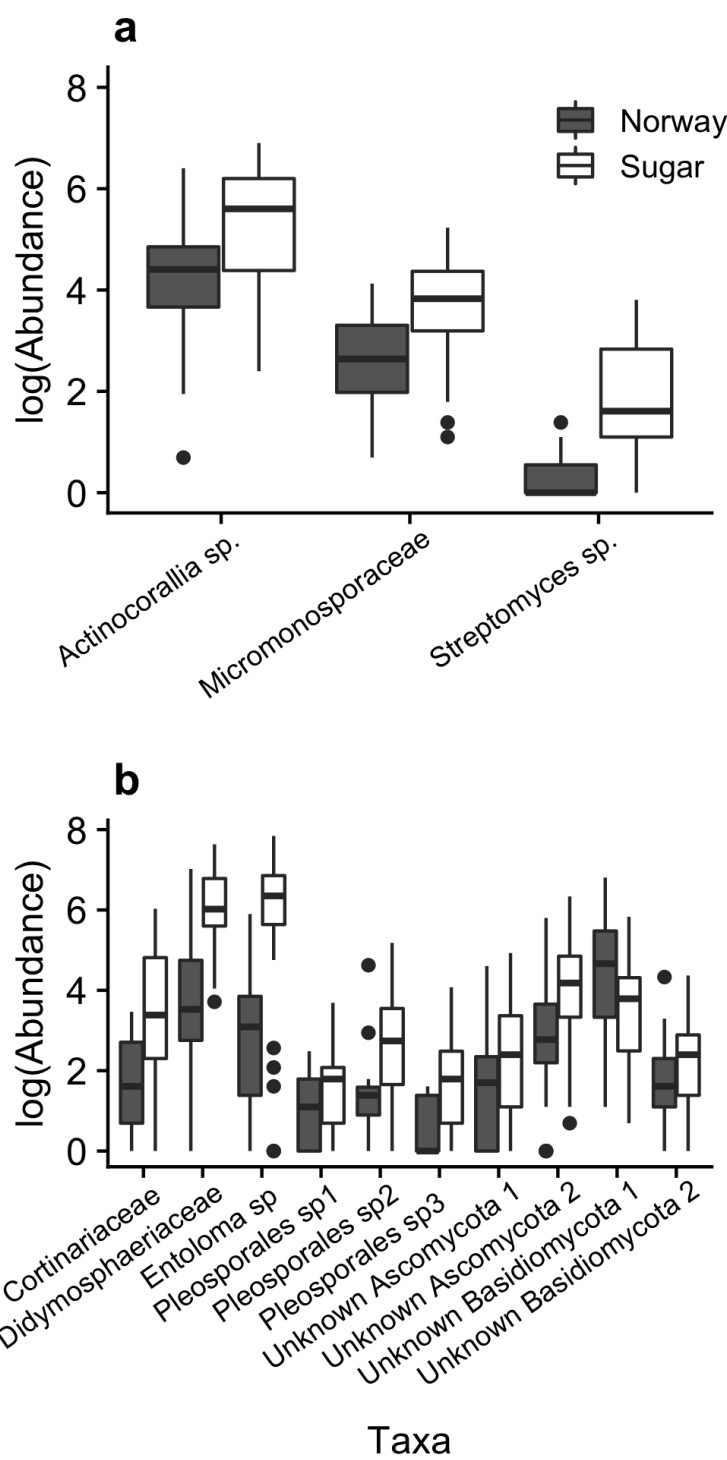

**Figure 3** Boxplots of median bacterial (A) and fungal (B) taxa that differed significantly in abundance among maple species (ANCOM, *P* < 0.05). Whiskers represent interquartile range (IQR) ×1.5.

## Quantification of differences in the richness of soil microbial communities on the roots of Norway and sugar maples

Bacterial OTU richness was significantly higher in Norway maple (288.23 $\pm$ SE 4.87) compared to sugar maple (251.74 $\pm$ SE 5.29, $t$-test; $P < 0.001$). The same pattern was seen with the fungi, with Norway maple having an OTU richness of 54.72 $\pm$ SE 2.76, and sugar maple 45.97 $\pm$ SE 2.36 ($t$-test; $P = 0.02$). AMF OTU richness did not differ between species. Norway maple had an OTU richness of 28.87 $\pm$ SE 1.58 and sugar maple AMF OTU richness was 27.94 $\pm$ SE 2.08 ($t$-test; $P = 0.72$).

## Quantification of differences in colonization rates of AMF and DSE on the roots of Norway and sugar maples

Overall, there was no significant effect of maple species on total AMF colonization (ANOVA; $F_{1,64} = 0.24$, $P = 0.62$). Total AMF colonization rates were equally high in both Norway maple and sugar maples with an average of 81.1% and 80%, respectively. However, date ($F_{7,64} = 22.09$, $P < 0.001$) and the interaction between species and date ($F_{7,64} = 2.94$, $P = 0.01$) was significant.

There was no significant effect of maple species on the abundance of arbuscules (ANOVA; $F_{1,64} = 0.206$, $P = 0.651$) and vesicles ($F_{1,64} = 0.029$, $P = 0.865$). The abundance of vesicles was low with an average of 2% and 2.1% for the Norway and sugar maples, respectively (Fig. 4A), while the abundance of arbuscules was similar and quite high in the Norway and sugar maples with an average abundance of 77.9% and 76.6%, respectively (Fig. 4B). However, the effect of date on the abundance of arbuscules and vesicles was significant ($F_{7,64} = 29.96$, $P = 0.006$ and $F_{7,64} = 9.16$, $P < 0.001$, respectively). There was also a significant interaction effect between maple species and date on the abundance of arbuscules ($F_{7,64} = 3.152$, $P = 0.006$) but not for vesicle abundance ($F_{7,79} = 1.28$, $P = 0.274$). No differences in vesicle and arbuscule colonization levels were detected between maples species at any of the eight sampling dates (Tukey post-hoc test, $P > 0.05$).

There was a significant effect of maple species and date on the colonization of dark septate endophytes (ANOVA; species: $F_{1,64} = 25.56$, $P < 0.001$; date: $F_{7,64} = 5.86$, $P < 0.001$), but the interaction term was not significant ($F_{7,64} = 2.10$, $P = 0.06$). The average rate of colonization of dark septate endophytes in the Norway maple and sugar maple was 16.5% and 33.4%, respectively (Fig. 4C). DSE were significantly more abundant in the sugar maple compared to Norway maple on August 28 and September 13 (Tukey post-hoc test, $P < 0.05$, Fig. 4C).

## Chemical analysis of the soil surrounding the Norway and Sugar maple roots

There were no significant differences in any of the measured soil chemical parameters among species and sampling dates (ANOVA tests; all $P > 0.05$). The only significant effect observed was the interaction term of species and date with C/N ratio ($P = 0.04$). C/N ratio decreased with time for Norway maple but slightly increased for sugar maple. ANOVA results and chemical analyses of the soil data set can be found in the Supplemental Information.

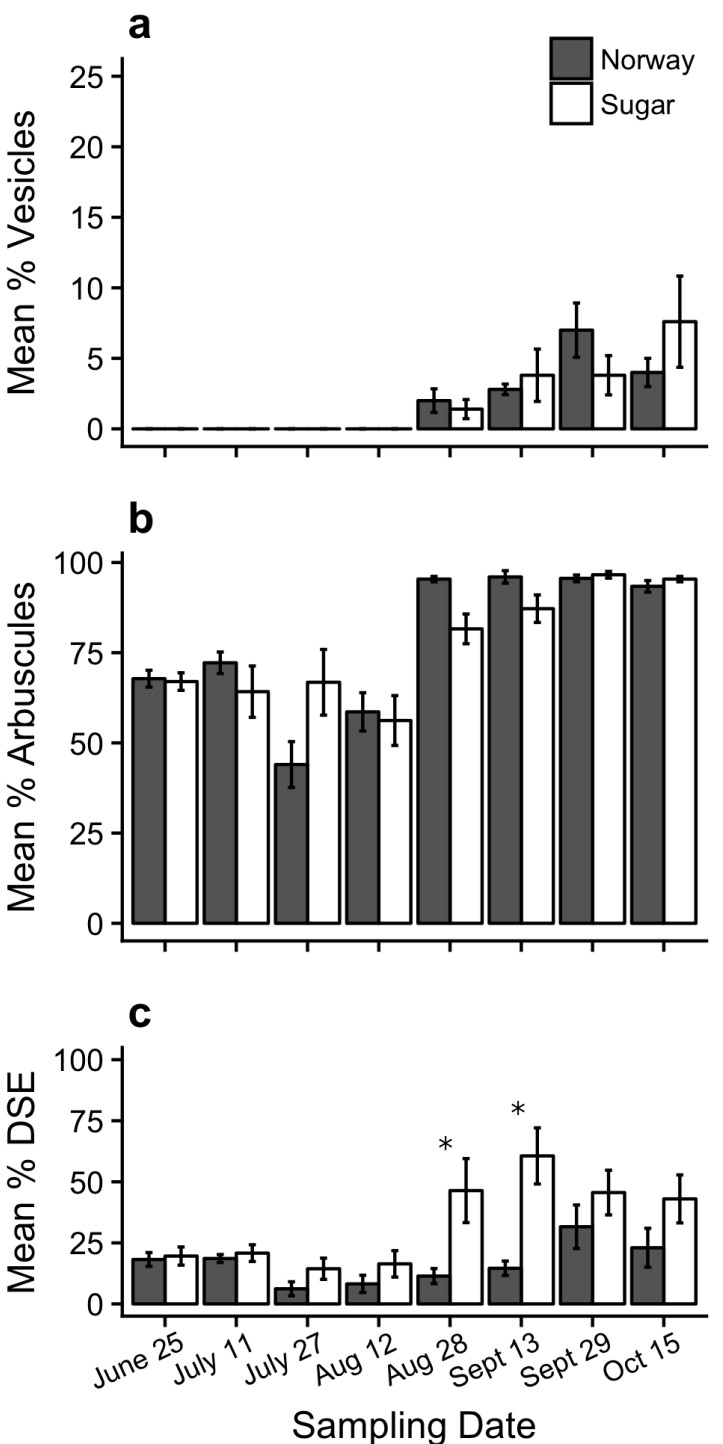

**Figure 4 Percent root colonization of mycorrhizal and dark septate endophytes in the Norway maple (Norway) and sugar maple (Sugar) throughout the sampling period (mean ± SE).** (A) Vesicular colonization, percent of root length containing mycorrhizal vesicles; (B) Arbuscular colonization, percent of root length containing arbuscules; (C) Dark septate endophyte colonization, percent of root length containing dark septate fungi. Symbols above bars indicate a significance difference ($P < 0.05$) between species at the specified sampling date based on the two-way ANOVA and Tukey post-hoc test.

## DISCUSSION

We examined the community composition of different groups of microbes simultaneously from the roots of Norway maple and sugar maple, and we show that soil microbial communities associate differently with host species. Our findings are consistent with the body of evidence showing that plant species identity can influence soil bacterial and fungal communities (*Haichar et al., 2008*; *Berg & Smalla, 2009*; *Leff et al., 2018*; *Linde et al., 2018*). We observed small but significant differences in microbial communities between these two co-occurring maple species albeit other work has suggested that closely related plants tend to share microbial communities (*Gilbert & Webb, 2007*; *Wehner et al., 2014*; *Johansen et al., 2016*), and time since invasion can diminish the impact of an invasive plant on the communities of soil microbes (*Lankau, 2011*).

Although host species has been shown to be an important factor shaping root associated microbes, numerous other factors also influence their community composition. The most abundant bacterial phyla in the maple roots were Proteobacteria and Actinobacteria. This is in agreement with a study by *Wallace, Laforest-Lapointe & Kembel (2018)* who examined microbes from roots and leaves of sugar maple seedings collected from a national park in eastern Quebec using similar methods. Proteobacteria and Actinobacteria were the most prevalent bacterial root endophytes in their samples, with Actinobacteria being an indicator taxon for the root compartment. However, there are distinct differences in the distribution of the dominant fungal taxa between our study and *Wallace, Laforest-Lapointe & Kembel (2018)*. The main fungal taxa we observed in the sugar maples were Ascomycota (52.7%) and Basidiomycota (41.7%), while the dominant fungal taxon reported by *Wallace, Laforest-Lapointe & Kembel (2018)* is Zygomycota (46.4%), followed by Ascomycota (40.1%), and they report relatively little Basidiomycota (12.4%). Some of the variation observed between the two studies may be attributed to time of sampling and site. *Wallace, Laforest-Lapointe & Kembel (2018)* only sampled once in the month of July and we sampled every two weeks from June to October for a total of 8 sampling periods. The multiple sampling periods may give a better overall representation of the fungi on the maple roots. However, date was not a significant factor explaining the variation of the fungal communities in our study (Table 1). Site differences may also be an important factor explaining for variation in fungal communities. An obvious site difference between our study and that of *Wallace, Laforest-Lapointe & Kembel (2018)* is that we sampled at a site where Norway maple has been present for over 50 years, with a high recruitment of Norway maple seedlings, therefore the soil microbes at our sampling location may be conditioned by its presence. The microbial assessment of maples at several locations would have to be compared in order to assess how much of the variation in fungal communities in sugar maples is a result of the presence of neighbouring Norway maples and other inter-site variations.

We detected higher bacterial and fungal richness on Norway maple compared to sugar maple, which may benefit the invasion process. Legumes that are able to associate with more bacterial species can successfully establish into new ranges compared to legumes that associate with a lower diversity of taxa (*Harrison et al., 2018*). *Toole et al. (2018)*

found higher fungal OTU richness in rhizosphere soils of Norway maple compared to sugar maple, however, they correlated fungal richness to plant weediness and not solely by the plant's invasive status. In their study, both the native and weedy red maple and invasive, weedy Norway maple had a higher fungal richness than native, non-weedy sugar maple and Japanese Maple (*A. palmatum* Thunb.), an introduced non-weedy plant, hence non-invasive species. More field sites and plant species would need to be sampled in order to determine if increased richness in root associated microbial communities is a common trend for range expanding species.

We observed high AMF colonization rates, in the roots of Norway maple, demonstrating its ability to effectively tap into the established native AMF network. We did not observe any differences in AMF community structure, colonization rates, or arbuscule to vesicle ratio between host species. This was somewhat contrary to what we predicted, but it does indicate that Norway maple is not hindered by a lack of suitable AMF in the introduced range. Moreover, it is possible that Norway maple derives more benefit from these associations with native AMF than does the sugar maple. *Klironomos (2003)* has shown that AMF can have different impacts on different plant species. An AMF survey by *Öpik et al. (2009)* showed that AMF specificity is more likely to occur at the level of ecological group rather than at the species level. The similarities in AMF communities between host species may be explained by the fact that we two maple species collected from a single location, observed no significant difference in soil chemistry surrounding the maple roots, and plants grown in close proximity can share mycorrhizal networks (*Van der Heijden & Horton, 2009*). Our results agree with other analyses of AMF in natural communities that have revealed that widely distributed AMF tend to associate with widely distributed habitat generalist plants (*Öpik et al., 2009*), and associating with generalist symbionts is proposed as a strategy for invasive plant species to avoid mutualism limitation in its novel range (*Moora et al., 2011*; *Nuñez & Dickie, 2014*).

Using morphological assessment of the root fungi, we found higher abundance of dark septate endophytes (DSE) on sugar maple compared to Norway maple. DSE are a group of ubiquitous ascomycete fungal root colonizers grouped together based on morphological characteristics (*Jumpponen & Trappe, 1998*; *Newsham, 2011*; *Mandyam & Jumpponen, 2015*). Despite the global distribution of DSE, their ecological role is still not well understood and ranges from beneficial to pathogenic (*Jumpponen, 2001*). DSE are not a taxon, therefore this range of ecological roles most likely results from the various taxa belonging to several orders of the phylum Ascomycota that are included in this group (*Jumpponen & Trappe, 1998*; *Newsham, 2011*; *Porras-Alfaro et al., 2014*). Of the nine fungal OTUs that had significantly higher abundances in sugar maple, four belong to the order Pleosporales, an order belonging to the DSE group (*Jumpponen & Trappe, 1998*). Although the ecological function of DSE is not well understood and historically they have predominately been considered benign, studies have reported negative effects of root DSE in tree species (*Wilcox & Wang, 1987*; *Tellenbach, Grünig & Sieber, 2011*; *Mayerhofer, Kernaghan & Harper, 2013*). Furthermore, *Tellenbach, Grünig & Sieber (2011)* not only showed that DSE had pathogenic effects but also showed that virulence was positively correlated with the extent of fungal colonization. Therefore, if DSE do negatively impact

maple species, we observed lower levels of DSE in Norway Maple roots compared to sugar maple, providing support for a possible reduced soil enemy effect in the invasive maple. We did observe that certain fungal OTUs were more abundant on the roots of sugar maple relative to Norway maple, unfortunately the FUNGuild results did not provide strong evidence for the pathogenicity of these fungi. Whether DSE do have negative effects on the maple species will need to be verified experimentally since we could not quantify the functional role of DSE in this particular study system.

## CONCLUSIONS

Here, we quantified the community composition of three microbial groups on the roots of Norway maple and sugar maple and showed that the bacterial and fungal communities varied between the two species, but no differences were found with AMF. Consequently, in order to obtain a full picture of the differences in root associated microbes between host species an assessment of the various microbial taxa is required. Specific microbes can have different impacts on different plant host and specific plant-microbe associations can affect plant coexistence (*Bever, 2003*); therefore, our observed patterns suggest a potential for belowground biotic interactions playing a role in Norway maple's invasion in North American temperate forests. Future work will be required to assess the functional capabilities of these microbes to determine their roles on the different maple species. The fact that we did observe differences in root microbes between the two species highlights the need to consider diverse microbial taxa when assessing the role of plant-soil interactions in the context of plant invasions.

## ACKNOWLEDGEMENTS

We gratefully acknowledge Victoria Pompa for her field assistance and for the morphological analysis of the roots. We also thank Evan Wisdom-Dawson, Richard Calvé, Alana Di Vito, Sean Di Paolo, who helped with the seedling collection and/or root washing, and Geneviève Lajoie for providing comments to an earlier version of this manuscript. We also would like to thank Mary-Ann Pavlik, Anne Godbout, Jim Fyles from the Morgan Arboretum, for their help at the Arboretum and permission to use the site.

### Funding

This research was supported with funding by the Fonds de Recherche du Québec—Nature et Technologies (FRQNT) (Tonia DeBellis), the Natural Sciences and Engineering Research Council of Canada (NSERC) (Steven W. Kembel and Jean-Philippe Lessard), and the Canada Research Chairs program (Steven W. Kembel). The funders had no role in study design, data collection and analysis, decision to publish, or preparation of the manuscript.

### Grant Disclosures

The following grant information was disclosed by the authors:

Fonds de Recherche du Québec—Nature et Technologies (FRQNT) (Tonia DeBellis).
Natural Sciences and Engineering Research Council of Canada (NSERC).
Canada Research Chairs program.

## Competing Interests

The authors declare there are no competing interests.

## Author Contributions

- Tonia DeBellis conceived and designed the experiments, performed the experiments, analyzed the data, contributed reagents/materials/analysis tools, prepared figures and/or tables, authored or reviewed drafts of the paper, approved the final draft.
- Steven W. Kembel and Jean-Philippe Lessard conceived and designed the experiments, contributed reagents/materials/analysis tools, authored or reviewed drafts of the paper, approved the final draft.

## DNA Deposition

The following information was supplied regarding the deposition of DNA sequences:

De Bellis, Tonia (2019): 2017 BACTERIAseqs.fna. figshare. Dataset. https://doi.org/10.6084/m9.figshare.7650104.v1

DeBellis, Tonia (2019): 2017 FUNGIseqs.fna. figshare. Dataset. https://doi.org/10.6084/m9.figshare.7650134.v1.

De Bellis, Tonia (2019): 2017 MYCORRHIZAEseqs.fna. Dataset. https://doi.org/10.6084/m9.figshare.7650137.v1.

DNA sequences are also available on the NCBI Sequence Read Archive: SRA accession: PRJNA553535.

## Data Availability

De Bellis, Tonia (2019): 2017-MYCORRHIZAEmetadata.txt. figshare. Dataset. https://doi.org/10.6084/m9.figshare.7650095.v1

De Bellis, Tonia (2019): 2017-FUNGImetadata.txt. figshare. Dataset. https://doi.org/10.6084/m9.figshare.7650080.v1

De Bellis, Tonia (2019). 2017-BACTERIAmetadata.txt. figshare. Dataset. https://doi.org/10.6084/m9.figshare.7650074.v2

De Bellis, Tonia (2019). 2017 Maple Soil chemistry.xlsx. figshare. Dataset. https://doi.org/10.6084/m9.figshare.7650143.v1.

## Supplemental Information

Supplemental information for this article can be found online at http://dx.doi.org/10.7717/peerj.7295#supplemental-information.

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
