# Peer review of "Shared mycorrhizae but distinct communities of other root-associated microbes on co-occurring native and invasive maples"

_PeerJ, doi:10.7717/peerj.7295_

## Round 0.1 · original submission · Major Revisions

The reviewers expressed similar concerns about the scope of inference given that this is a correlative study of only one species pair at one site. Your resubmission should limit the introduction and discussion to the scope of the objectives as described by the reviewers.

Reviewer 1 ·

Basic reporting

The study compares and contrasts patterns of root associated microbes of seedlings of two co-occurring maple trees (native and non-native comparison). Though the study is descriptive (i.e. compares associational patterns), the authors spend a great deal of space attempting to interpret the likely functional importance of these patterns. I think they should devote most of their lines to describing the patterns and not trying to interpret functional importance (e.g. how microbes might influence Norway maple's invasion). For example, I noted that their conclusion statement includes much too much on functional interpretations which are beyond the scope of the study. There are also concerns that the maple-to-maple variation might be typical of tree-to-tree variation among resident tree species.

Experimental design

A general description of the sampling design is provided (L185-91) but there is not a clear description of total number of collected seedlings per species. Did you collect just one of maple spp. seedling specimen per transect per period (n=1)?

A potential limitation is that don’t we expect species to species variation in compositional associations? In other words, is the variation between the native vs. non-native maples actually greater than simply the variation between sugar maple and an average other co-occurring resident woody species? By having only one native and one non-native species, it is difficult to interpret whether the variation in associational similarity is typical for differences between any two species or something that is somehow special/atypical.

Validity of the findings

See reply to #1 & #2 above.

Additional comments

Abstract- Discussion points are highly speculative (e.g. L45-6). Your study describes patterns of just two host species and includes no experimental manipulations that are necessary for comparing soil biota effects differences for the two hosts.

L88 I think the Klironomos citation is technically wrong. John’s study did various things. One was to compare abundant exotics to rare natives, which found differences in effects. But those differences were confounded by plant abundance. Anacker et al. (2014) re-analyzed John’s data (large community dataset) and found no differences between feedbacks of natives and exotics when controlling for abundance. In other words, rare plants tended to have negative feedbacks and abundant plants tended to have neutral to positive feedbacks regardless of invasive status.

L105 It seems you might want to draw on the plant pathology literature here. I don’t think these papers actually directly isolated and measured pathogenicity.

L110-3 Del- “Klironomos (2002) showed that invasive plant species in eastern Canada had neutral feedback to soil communities while a negative feedback was observed in native plant species suggesting that invading plant species may have escaped from their natural soil pathogens.” See comment above for L88.

L110-6 I think you could have drawn on better literature for these comparisons (e.g. Reinhart and Callaway 2004, Reinhart et al. 2010, Halbritter et al. 2012). Though you may want to limit content on the functional importance of microbes since this is not the focus of your study...

L122 A couple of papers have reported the opposite (Reinhart and Anacker 2014, Veresoglou and Rillig 2014).

L141 “Rinehart” check spelling

L151 try- and/or

L167 “deceased”

L170 “cooccurring”

L199 “microtubes tubes”

L252 “demuliplexed”

L323 “comminutes”

L353 This isn’t very compelling evidence for differences in enemy impacts between the two maples.

L396 “low in with”

L422-5 These sentences are on the functional importance of microbes but this was not directly measured in your study. Shouldn’t your ms stay focused on differences in microbe associational patterns?

L428 Your comparison is extremely limited. What I mean is that you compare one native maple to one non-native maple. Might you see the same differences if you compared co-occurring red and sugar maple? In other words, shouldn’t we assume that some associational variation can be detected even among closely related species? Yet, this does not tell us about popn dynamics of the two species… The comparisons are so limited (one native sp. vs. one non-native sp.) that it is difficult to gauge if such small compositional differences (e.g. ordinations with lots of overlap) truly tell us much about invasion processes.

L440 Highly speculative.

L443 try- “compared to native maples”

L459 detrimental to what (tree or microbes)? Richness topic is extremely vague and not necessarily grounded in theory. For example, I don’t think disease outbreaks are typically related to microbial richness as opposed to pathogen prevalence/presence. While I recognize that a community may contain microbes that also protect the host from the pathogen, this is not necessarily a direct attribute/effect of richness. One strong pathogen (or antipathogen) could have huge impacts that are largely independent of richness.

L470 try “Acer” or maple

L505 Conclusion statement is heavily focused on perceived importance of root association patterns which is beyond the scope of the study. It seems it should summarize the main patterns and minimize interpreting these patterns.

Literature cited

Anacker, B. L., J. N. Klironomos, H. Maherali, K. O. Reinhart, and S. Y. Strauss. 2014. Phylogenetic conservatism in plant-soil feedback and its implications for plant abundance. Ecology Letters 17:1613-1621.
Halbritter, A. H., G. C. Carroll, S. Güsewell, and B. A. Roy. 2012. Testing assumptions of the enemy release hypothesis: generalist versus specialist enemies of the grass Brachypodium sylvaticum. Mycologia 104:34-44.
Reinhart, K. O., and B. L. Anacker. 2014. More closely related plants have more distinct mycorrhizal communities. AoB Plants 2014:doi: 10.1093/aobpla/plu1051.
Reinhart, K. O., and R. M. Callaway. 2004. Soil biota facilitate exotic Acer invasion in Europe and North America. Ecological Applications 14:1737-1745.
Reinhart, K. O., T. Tytgat, W. H. Van der Putten, and K. Clay. 2010. Virulence of soil-borne pathogens and invasion by Prunus serotina. New Phytologist 186:484-495.
Veresoglou, S. D., and M. C. Rillig. 2014. Do closely related plants host similar arbuscular mycorrhizal fungal communities? A meta-analysis. Plant and Soil 377:395-406.

Reviewer 2 ·

Basic reporting

The manuscript is well written and presented in an easily understandable language. The introduction list several mechanisms and theories regarding the role of microorganisms for invasive plants, but this is not really the aim of the manuscript, as the aim only is to see if there are differences in the microbial communities of invasive and native plants.

Experimental design

I have serious concerns regarding the experimental design. The survey is conducted within one 40 x 60m plot. In that sense, it is pseudo replicated, as more sites should have been studied to conclude that the two plant species differ in their microbial communities. Moreover, the host species only explained 3.6% of the bacterial variation and 8.4% for the fungal communities, indicating that environmental variables could be more important. This could be different in another area.

Validity of the findings

I do not agree with the conclusion that the difference in communities are due to the invasiveness of the Norway maple. Such differences could probably also be found between two native plant species. It would be highly unlikely that the two species should have exactly the same communities given their difference in growth pattern. If the differences are due to the invasive characteristics of the Norway maple, this should also been seen at other plots

---

## Round 0.2 · Minor Revisions

The previous reviewers were not available to re-review the manuscript, therefore I reviewed it myself. I believe most of the reviewers concerns were addressed in the revision, but I have a few minor revisions to suggest.

Line 135. Could you please clarify what is meant by plant identity? Is this species and genotype or are these studies assessing one or the other? Also I suggest “Plant identity can be an important factor…” because it sounds like it is not always “the” important factor.

Lines 545 and 727. I would prefer you not claim to be the first to do this work. PeerJ does not make editorial decisions based on novelty, so it is not needed, and I cannot easily validate this statement.

First paragraph of discussion. The sentence from lines 547 to 551 should be broken into two, one about your results and a second on explanations for the small difference between communities. It is hard to follow as is. Also I think the statement about the history of the field site should go earlier in the paragraph to provide context to the results. Currently it seems out of place between two comparisons to other literature.

Lines 589-595. I’m not convinced that these studies are relevant. One is comparing invaded and native ranges and the other is comparing invaded to non-invaded. Neither of these comparisons are present in your study. I find these sentences confusing, or worse, misleading.

Minor corrections:
Line 53 communities
Line 112 remove “and”
Line 145 plants (again, is this closely related plant species or plant genotypes?)

---

## Round 0.3 · accepted · Accept

Thank you for your edits!